# Psychological Effects of Nasogastric Tube (NGT) in Patients with Anorexia Nervosa: A Systematic Review

**DOI:** 10.3390/nu16142316

**Published:** 2024-07-18

**Authors:** Federico Amianto, Tomaso Oliaro, Francesca Righettoni, Chiara Davico, Daniele Marcotulli, Benedetto Vitiello

**Affiliations:** 1Neurosciences Department, University of Torino, Via Cherasco 15, 10126 Turin, Italy; 2Department of Pathology and Care of the Child, AOU Città della Salute e della Scienza di Torino, OIRM, 10126 Turin, Italy; tomaso.oliaro@unito.it (T.O.); francesca.righett964@edu.unito.it (F.R.); chiara.davico@unito.it (C.D.); daniele.marcotulli@unito.it (D.M.); benedetto.vitiello@unito.it (B.V.); 3Department of Public Health and Pediatric Sciences, University of Turin, 10126 Turin, Italy

**Keywords:** anorexia nervosa, nasogastric tube feeding, NGT, inpatient treatment, psychological consequences of treatment

## Abstract

Aim: After the COVID-19 pandemic, the need for intensive nutritional care in patients affected with anorexia nervosa (AN) increased. The use of NGT was often used to overcome renutrition difficulties. This systematic review explores the evidence concerning the psychological effects of an enteral nasogastric tube (NGT) feeding on patients with AN. Methods: A systematic review following PRISMA guidelines was conducted on electronic databases, including papers from January 2010 to December 2023. The keywords used combined anorexia nervosa, NGT, nasogastric tube, and tube feeding, with MeSH terms. No language limit was imposed. Reviews were excluded from the search. Results: A total of 241 studies matched the keywords. Nevertheless, 236 studies were excluded from the review because they did not match the inclusion criteria. A total of six studies met the inclusion criteria. Of these, three studies were case series, one was a quantitative study of follow up and one was a qualitative exploratory study. The included studies described the hospitalization of patients with AN treated with a nasogastric tube; among these, only one study focused directly on the psychological correlates of nasogastric tube treatment using interviews with patients and medical staff. Included studies suggest that NGT feeding, even if faced in the first instance with prejudices and fears by patients, parents, and staff, is useful not only for weight increase in treatment-resistant patients with AN, but also alleviates their stress from feeding and, in general, it is psychologically well tolerated. Nevertheless, recent in-depth research on the issue is lacking and the existing has a low methodological quality; thus, many psychological effects of NGT application remain underexplored. Conclusions: Although the results suggest good psychological tolerance of the device, the limited data available recommend that more attention should be addressed by the researchers to the psychological consequences of the use of NGT in the treatment of AN since it is a nutrition disorder with prominent psychological roots. Further studies are needed.

## 1. Introduction

Anorexia nervosa (AN), according to the DSM 5, is a psychiatric pathology characterized by excessive restriction of calories consumed, intense fear of gaining weight, and a distorted body image [1]. In the United States, the lifetime prevalence rate is estimated at 0.3% in men and 0.9% in women [2]. A peak incidence has been identified between the ages of 15 and 19 [3]. An increase in incidence has been observed over the last 20 years in every age group [3]. After the COVID-19 pandemic, a new important increase (up to 67%) of this eating disorder has been observed [4] and a greater increase in severe cases with high psychiatric comorbidity and a need for inpatient treatment (104%) was reported [5]. Moreover, the prevalence is estimated to be higher than that reported, since numerous cases do not reach medical observation [3]. Finally, mortality for this pathology is the highest among psychiatric diseases with a rate of 5.86 [3].

The treatment of AN is complex and multidisciplinary, it encompasses psychiatric, nutritional, and psychological approaches [6,7]. Early management by the medical team is mandatory and has proven to be a favorable predictive factor for recovery [6]. However, the drug therapy has demonstrated poor effectiveness and there are currently no data in the literature confirming that they are effective for the treatment of the disease [6,7,8]. Drug therapies are mainly used as symptomatic treatments. Some of the more specific therapies for AN are atypical antipsychotics and are used to reduce obsessive behavior, agitation, motorreha, and anxiety that develop following refeeding, but sometimes some of them [e.g., olanzapine] are also administered because they can increase hunger [8]. Generally, the use of antipsychotics in AN is associated with antidepressant and anxiolytic drugs that address mood and anxiety comorbidity [7].

Regardless of the therapeutic approach, it is essential for the treatment of AN to establish a good therapeutic alliance with the patient to overcome the initial resistance to the treatment [9]. For some patients who are in the acute phase and who are not compliant with dietary therapy or who are in a critical clinical situation, it may be necessary to provide for their caloric and nutritional needs through the nasogastric tube (NGT). The introduction of NGT in patients with anorexia nervosa (AN) is typically considered in specific clinical scenarios [10,11]:Severe Malnutrition and Weight Loss: When patients present with severe malnutrition or have lost a significant amount of weight (typically more than 15–20% of their body weight), nutritional support through an NGT may be necessary to prevent further health deterioration and to stabilize the patient’s condition. This is especially critical when oral intake is insufficient to meet the caloric and nutritional needs required for recovery.Failure of Oral Refeeding: If patients are non-compliant with dietary therapy or oral refeeding plans, an NGT can ensure that they receive the necessary nutrients. This approach helps in overcoming the resistance to eating commonly observed in AN, thereby facilitating weight gain and nutritional rehabilitation.Critical Clinical Condition: In cases where patients are in a life-threatening condition due to complications such as electrolyte imbalances, cardiovascular instability, or severe organ dysfunction, immediate nutritional intervention via NGT is crucial to support vital functions and initiate the recovery process.

Overall, the decision to introduce an NGT is made on a case-by-case basis, considering the patient’s medical condition, nutritional status, and the risk–benefit ratio of the procedure. The goal is to stabilize the patient, ensure adequate nutrition, and transition back to oral feeding as their condition improves [9,12]. The NGT consists of a thin tube that passes through the nose, reaches the stomach, and serves the infusion of nutritional food [13]. There is a low risk for complications associated with the introduction and maintenance of the NGT and the nursing staff receives specific training for management and correct positioning [11]. There is the possibility of using the NGT with different infusion methods; however, the most used are the nocturnal infusion and the diurnal bolus [13].

Several studies in the literature have highlighted the benefits of the nasogastric tube in terms of weight recovery, improvement of internal parameters, and risk of refeeding syndrome [14,15]; however, the literature is still lacking regarding the psychological repercussions of NGT forced feeding in patients affected by AN. The study from Parker et al. [16] demonstrates that malnourished adolescents hospitalized with restrictive eating disorders can safely begin a nutritional rehabilitation program with an average intake of 2611.7 kcal per day. This aggressive approach led to significant initial weight gain rates exceeding current recommendations. Prophylactic phosphate supplementation effectively prevented hypophosphatemia, a key concern during refeeding. This study underscores the safety and efficacy of early phosphate supplementation in preventing refeeding syndrome. This study suggests that rapid refeeding, supported by NG feeding when necessary, can safely achieve quick weight restoration, potentially reducing hospital stays and facilitating psychological therapy.

The study by Marchili et al. [13] evidences that patients treated with NGT are generally more severe but can recover more quickly and with good stabilization of the results. It examined the impact of nasogastric tube (NGT) feeding on 315 predominantly female (89.2%) patients with anorexia nervosa (AN) in a hospital setting with a mean age of 14.4 years. In total, 32.1% of these patients required NGT feeding due to severe caloric intake reduction. NGT-treated patients exhibited more severe clinical profiles, including lower BMI and higher rates of inpatient psychotropic therapy (IPDT) use. They required a longer hospital stay (30 ± 11 days) compared to non-NGT patients (16 ± 9 days). Interestingly, NGT-treated patients had longer remission periods (8.2 months) compared to those without NGT (3 months), indicating that NGT plays a positive long-term role in nutritional recovery. This was associated with better compliance with oral feeding and improved psychological states, leading to earlier weight restoration and extended remission periods. The study also found that NGT had similar impacts on increased BMI as IPDT, but suggested that NGT might be more suitable for pediatric patients without psychiatric comorbidities due to the possible side effects and resistance associated with drug treatments.

The study by Rigaud et al. [17] evidenced that a nutritional strategy based on a low-sodium diet may be useful in reducing side effects related to NGT use. It focused on the refeeding protocols of 218 malnourished AN patients with NGT. The study compared two refeeding protocols: a normal-sodium diet and a low-sodium diet. The first group (42 patients) experienced significant weight gain but also higher incidences of leg edema (21%) in those with a BMI < 15 kg/m^2^. In contrast, the 176 patients on a low-sodium diet showed a more moderate weight gain and lower incidences of leg edema (6%). The study reported a high tolerance for tube feeding, with no severe side effects, refeeding syndrome, or metabolic issues. Sinusitis was a minor complication, affecting 6% of participants. During the study, a slight decrease in albumin and hematocrit was observed after two months. Over two months, patients on the low-sodium diet achieved substantial weight gain, with a medium BMI increasing from 12.6 to 17.6 kg/m^2^. A notable finding was a BMI plateau between 15 and 16 kg/m^2^, suggesting an adaptation phase in the body’s response to refeeding. The study concluded that a low-sodium diet helps manage water retention more effectively, providing a more accurate measure of weight gain and reducing patient anxiety.

The study from Puccoli et al. [18] evidenced that the NGT might offer benefits in terms of treatment compliance and nutritional status stabilization. It was conducted on 79 patients with AN, who were 98.7% female, with a mean age of 14.6 years, 98.7% with the restrictive subtype of AN, and with an average untreated illness duration of 14.8 months. Comorbid conditions included obsessive–compulsive disorder, major depressive disorder, and panic disorder. All patients received NGT feeding, and 86.0% also received atypical antipsychotics (AAP). The participants were divided into five groups based on the timing of AAP and NGT treatments. The most frequently used AAP was olanzapine. Early introduction of both AAP and NGT significantly reduced the length of hospital stay. The early AAP NGT group had a significantly shorter stay (81.3 ± 31.4 days) compared to groups with later AAP introduction. The findings suggest that early intervention with AAP and NGT can enhance recovery outcomes by improving compliance, reducing agitation, and addressing body image distortions. The study emphasized the importance of timing in treatment interventions, proposing that early introduction of both AAP and NGT could lead to better physical and psychological recovery in AN patients.

In their complex, the studies conducted on nutritional consequences of the NGT refeeding evidence its usefulness in the short and medium term in reaching more rapid and sustained nutritional well-being, the possibility to reduce its side effects with specific nutritional approaches and adequate timing with AAP use, and the positive consequences on behavioral problems and compliance [13,16,17,18].

In the years preceding 2010, some studies have considered the psychological consequences of the use of NGT in patients affected with anorexia nervosa. In particular, the study by Rigaud et al. [19] included eighty-one adult patients suffering from AN. From them, 25 patients suffered from the binge/purging type of AN and 56 from the restrictive one. All patients fulfilled the AN criteria of the DSM IV; 98.4% were amenorrhoeic and 36% had physical hyperactivity of at least three hours per day. This study evaluated how nutrition through NGT gives better results in terms of body weight gain, rate of weight gain, fat-free mass gain, total energy intake, and decrease in purges/binges episodes more rapidly than other control cases and how the perception of forced feeding changes through time. The last was assessed through self-administered questionnaires filled out weekly for four weeks that investigated whether the use of NGT increases the fear of eating, whether NGT is helpful for weight recovery, and if NGT prevents people from eating, using a 6-point Likert-scale; after four weeks of treatment, the impression regarding the NGT feeding changed and the majority of patients stated that NGT was useful for recovery and helped them to eat. Moreover, as a relevant psychological outcome, it has been evidenced that the NGT did not increase the level of anxiety as compared to the group of control cases.

In a study by Zuercher et al. (2003) [20], all patients diagnosed with AN (381 patients total: 155 with NGT and 226 without NGT) were given questionnaires to complete: the EDI-2 (self-administered test evaluating the food symptoms) to fill out at the moment of hospitalization and right before hospital discharge, and the PSQ (self-administered test evaluating the patient’s level of satisfaction) to fill out only once at the moment of hospital discharge. No significant differences in the general satisfaction of the treatment were found between the group with NGT and the group without NGT.

In general, the studies on the psychological consequences of NGT use evidenced that AN patients’ negative impression of the NGT use changed over time and their levels of anxiety did not increase, while their satisfaction was comparable to the treatment without NGT refeeding [19,20].

Nevertheless, even though the results were encouraging for the use of NGT in the treatment of AN from a psychological point of view, the overall quality of the information is limited, the number of explored patients was sometimes scarce, no validated instrument was adopted to assess NGT effects, no report considered the medium- and long-term psychological consequences of the NGT use, and the possible problems related to psychiatric comorbidity or personality traits. Finally, no study explored the implications of NGT treatment for family and medical staff. In this study, we aimed to assess the research advances in the last fourteen years (since the last comprehensive review) concerning the psychological effects of NGT application in the treatment of AN by exploring the literature regarding the use of NGT exploring pertinent literature from 2010 up to the present.

## 2. Review Methodology

### 2.1. Search Strategy

The research question for this study was developed following the PICO (Population, Indicator, Comparator, and Outcome) criteria to create an effective research strategy.

This review applied the PRISMA (Preferred Reporting Items for Systematic Reviews and Meta-Analysis) guidelines.

The literature search was conducted on electronic databases, from January 2010 to December 2023, as follows: WorldCat; MEDLINE^®^/PubMed^®^; Scopus; Directory of Open Access Journals; and Wiley Online Library. The keywords used combined anorexia nervosa, NGT, nasogastric tube, tube feeding, and enteral feeding with MeSH terms. The keywords used were combined as follows: (a) NGT and anorexia nervosa; (b) nasogastric tube and anorexia nervosa; (c) tube feeding and anorexia nervosa; and (d) enteral feeding and anorexia nervosa.

The included databases were all those available online for the international medical literature research of our academic institution, and none were excluded.

### 2.2. Inclusion and Exclusion Criteria

The aim of this review was to explore the psychological effects of the use of the nasogastric tube on patients diagnosed with anorexia nervosa.

The included studies had to report the psychological effects or outcomes, and/or patient impressions concerning the use of the nasogastric tube.

We chose not to narrow the search due to the scarcity of articles on the topic.

We therefore decided not to limit the search to English language articles. It also did not impose any restrictions on the type of study, with the only exclusion being reviews. The authors are aware of the possible difficulties derived from the need for translation and/or interpretation of the relevant non-English articles, but they accepted the challenge with the aim of producing a complete review making use of payment translation services and free translation services also based on AI. Furthermore, the authors recognize the value that systematic reviews can provide in offering summarized insights. However, they made a deliberate decision to exclude such reviews to generate original content and adhere to the timeframe of their research. No restrictions were imposed regarding the study design, with the inclusion of qualitative and quantitative studies. The choice of including studies with different primary aims and methodological approaches is justified by the scarce number of available research on the topic but also because of their possible contribution to the definition of psychological correlates of the NGT use in AN treatment from different perspectives.

The database search was complemented by the review of studies cited in relevant articles (back citations).

To be eligible for the inclusion criteria, participants had to be adolescents or adults diagnosed with AN according to the DSM 5 criteria. No selection was made in terms of the gender of participants.

Studies that only reported the clinical effects of the NGT (weekly weight gain, total weight gain, discharge weight, and discharge body mass index) or the patient’s tolerance exclusively from a physical point of view were excluded.

### 2.3. Study Selection

After downloading the search results, inclusion and exclusion criteria were applied to titles and abstracts by two reviewers independently.

Articles that passed this first review were retrieved in full and the full text was reviewed by two of the review authors to determine eligibility for inclusion of the articles for the final analysis.

Figure 1 describes the flowchart of the papers’ selection. The research on the mentioned databases selected 241 studies meshed with the inclusion criteria. In total, 62 duplicate papers and 37 book chapters were eliminated, reaching the number of 142 studies for content analysis. In the first step, 27 papers were removed because they did not include NGT treatment, and another 13 were removed as they did not include a reference to the psychological aspects of the NGT treatment. In the second step, 102 studies were selected but 16 were removed because they did not include patients affected with AN, and, furthermore, 81 studies were removed because they did not address the psychological tolerability of the NGT. A total of 6 papers were therefore included in the present review, which addresses the psychological perception of the NGT treatment by AN participants.

### 2.4. Data Extraction

The research questions and data extraction were guided by the PICO framework. The study population included all patients with AN according to DSM5 criteria who were fed by a nasogastric tube. There were no age or gender restrictions. The intervention we analyzed was enteral feeding via NGT, and our primary outcome measure was the psychological tolerance of the patients. The secondary outcome measure was the impact of this feeding practice on the healthcare team and caregivers.

### 2.5. Quality Assessment

Due to the observational design of the majority of the analyzed studies, it was not possible to use a formal evaluation instrument [21]. The limitations of each study are described in the next section.

## 3. Results

As shown in the PRISMA flowchart (Figure 1), 102 articles were reviewed in full text and 96 were excluded. Among these, 15 did not concern the diagnosis of AN (15.6%) and 81 did not address the psychological tolerance of treatment (83.5%). A total of six studies (5.9% of the total) met the inclusion criteria for the present review. Among these, three studies were case series, one was a follow-up quantitative study, one was a qualitative exploratory study, and one was a prospective cohort study. Overall, the studies of this review collected the experiences of 305 patients with AN. Among them, 258 were female (76.7%), 17 were male (8.4%), and 30 were reported as a gender different from those they were born with (14.9%). The average age of the sample is 18.1 years. Table 1 summarizes the main characteristics of the studies and Table 2 shows the main data on patients included in the studies.

### 3.1. Analytic Description of the Studies

Bayes and Madden (2011) [22] evidenced that the cooperation of AN patients is essential to the success of NGT and that their contradictory attitudes and ambivalence about it should be eventually addressed with drug therapies. The study collected the experiences of 10 inpatients in two of Sydney’s children’s hospitals for eating disorders. A particularity of this study is the all-male population, aged 10 to 14 years. The study reconstructed the social and family context in which patients lived and the patterns of eating disorder onset. During hospitalization, which lasted an average of 36.3 days, six patients (60%) required NGT feeding. In the discussion, the authors did not address directly the psychological consequences of the NGT introduction and its associated feelings but commented that optimal conditions for introducing the NGT would involve patient cooperation, and they suggested that the administration of drugs such as atypical antipsychotics is useful in order to alleviate agitation during the insertion process. To underline the importance of psychological aspects, the authors cited two articles: Halse et al. (2005) [28] and Neiderman et al. (2001) [29]. In the first study, contradictory meanings were highlighted regarding the use of the tube: “[I] feel better [after the insertion of the NGT –note of the authors] even though I didn’t want it”. And the second study recommended the introduction of NGT feeding with the patients’ collaboration. Limitations of this study include its retrospective design, which may have impaired the precise recollection of the patients, a post hoc reconstruction of their history biased by its evolution, and the incomplete nature of information obtained from patient files. The small number of participants limits the generalizability of the findings. This study only looked at boys with AN or EDNOS admitted to hospital because of medical instability, which would represent only a portion of the total number of boys with eating disorders in the population. The authors concluded the discussion of their article by pointing out the lack of a specific approach in the treatment of weight recovery.

The study by Rigaud et al. (2011) [23] evidenced that for patients with severe bulimia nervosa (BN) or anorexia nervosa (AN) who do not respond to conventional treatments like cognitive behavioral therapy (CBT) or antidepressants, NGT feeding has shown promising results. This randomized controlled trial involved 103 adult outpatients suffering from binge–purge (BP) behaviors, with 52 patients receiving NGT feeding combined with CBT (TF group) and 51 receiving CBT alone (CBT group). The study found that NGT feeding significantly increased abstinence from BP episodes compared to CBT alone, with 81% of the TF group achieving abstinence at eight weeks versus 38% of the CBT group. Additionally, the TF group showed greater improvements in nutritional status, including increases in fat-free mass and key biological markers, as well as better quality of life and reduced anxiety and depression levels. These results were maintained for up to 12 months in a significant portion of patients, demonstrating the potential effectiveness of NGT feeding as part of a comprehensive treatment approach for refractory BP disorders. Quality of Life (QOL): QOL improvement assessed through SF-36 was significantly greater in the TF group compared to the patients who received only CBT. Mental and Well-being Scores. The TF group showed significantly greater improvements compared to the CBT group across all comparisons. BDI Score: Improvement in BDI score was more pronounced in the NGT group compared to the CBT group. HAM-A Score: The TF group had lower HAM-A scores compared to the CBT group, with differences persisting at the 12-month follow-up. EDI Score: The TF group showed a significant reduction in EDI score compared to the CBT group. The study’s strength was the coupling of the NGT treatment with a psychological approach and the objective assessment of self-perceived QOL, eating psychopathology, depression and anxiety; nevertheless, the assessment of patients’ perception of NGT treatment was not carried out.

Blikshavn et al. (2020) [24] conducted the only study in the current review examining the effects of physical restraint associated with NGT use. It underscores how the use of NGT may pose significant challenges for AN patients, to the extent that physical restraint is required. The study was a quantitative study on 58 patients with the aim of describing the use of physical restraint during enteral nutrition in patients hospitalized for AN in a pediatric center and testing the association between weight gain and the use of restraint and the 5-year outcome of these patients. In total, 89.5% of the patients were female and 10.5% were male; the average age was therefore 15.9 years, and 100% of the patients had a diagnosis of AN. In this study, there was a need for physical restraint on 38 patients (66%) only during the day and never during the night time; among the 170 restraint episodes with a known cause, the majority, comprising 54% (*n* = 109), were due to the necessity of administering nutritional treatment, while 28% (*n* = 56) were instances where restraint was utilized to prevent serious self-harm. These two categories combined constituted 97% (*n* = 165) of all recorded episodes; although this was not the purpose of the study, this suggests that nutrition via NGT may be psychologically challenging to endure, yet essential for the treatment of these patients. The main limitation of the current study is the small number of patients in the group of interest, leading to a high risk of type II errors. Moreover, the study only reports and analyzes the restraint episodes without an assessment of patients’ personality and psychopathology features, as far as psychiatric comorbidity. No objective assessment of patients’ perception of NGT treatment was carried out.

Falcoski et al. (2021) [25] explore the complex psychological and practical challenges of NGT feeding under restraint in adolescent eating disorder treatment and, as in the previous study, suggest managing them using psychotropic drugs. They reported three cases of inpatients with a diagnosis of (typical or atypical) anorexia nervosa fed by NGT under restraint. The patients described in this study are between 11 and 14 years old.

For each case, a table summarized the nutritional characteristics of the feeding plan and comments from the health staff regarding the modality of intervention. For the first inpatient, for example, the presence of “one person facilitated hand holding to reduce distress and provide reassurance” is reported. Further comments were the following: “Verbal support/just some nausea after”, “Restrained × 2 staff”, and “Hand support”.

The cases highlight various approaches to managing feeding volumes, rates, and caloric intake to mitigate risks like nausea, vomiting, dumping syndrome, and refeeding syndrome. Case 1 involved a patient receiving bolus feeds up to 1650 kcal, carefully managed to prevent refeeding syndrome without adverse effects. In Case 2, a high-calorie feed (2250 kcal) was administered via Ensure Compact and Calogen to meet nutritional requirements, with mild nausea managed post-feed. Case 3 focused on a patient with compulsive exercise patterns, requiring sustained NGT feeding over two months, gradually increasing to tolerate feeds exceeding 1000 mL. Psychological aspects were fundamental, with patients receiving support to accept and manage NGT feeding under restraint. Medications like olanzapine and lorazepam were used to alleviate anxiety and facilitate compliance, with no compromise of feeding tolerance. The decision to retain NGT was made in collaboration with the patients, ensuring safety without causing nasal trauma through frequent reinsertions. However, patient and parental perceptions varied, with some viewing NGT feeding as necessary for treatment progress, while others found it distressing yet unavoidable. While effective in nutritional rehabilitation, it requires careful management to address psychological trauma and ensure patients’ safety and compliance. The authors discussed why there were such few episodes of NGT nutrition under restraint, giving credit to a combination of factors: greater patient awareness, greater knowledge of treatment procedures, and support from staff mainly via hand holding and verbal support. The authors therefore questioned the potential psychological trauma that NGT treatment may carry in some patients. They cited studies that found that the majority of patients and parents considered the use of NGT as a necessary and essential practice, that NGT alleviated the guilt of introducing calories in patients with intrusive anorexic thoughts [29], and that enteral feeding was safe and well tolerated [30]. This study has some limitations: the small number of featured patients, which reflects the rarity of situations requiring this type of intervention, and that reduces the generalizability of the findings to the whole AN population. Moreover, different settings have various policies, such as general pediatric hospitals versus specialist eating disorder units. These diversities may generate a treatment bias and also a different relational approach to patients who may have influenced the psychological outcome of the treatment. Furthermore, the study did not incorporate the perspectives of patients or parents regarding potential future outcomes.

Matthews-Rensch et al. (2022) [26] explore the point of view of patients and staff concerning the use of NGT for feeding AN patients and report positive and negative perceptions of both. The authors investigated the acceptability of NGT in eating disorders treatment through semi-structured interviews conducted between June and November 2019. The study is particularly appreciable because it doubled the point of view on the issue by interviewing not only patients but also medical staff (medical *n* = 5, nursing *n* = 5, dietitians *n* = 2; with a median of 8.5 years of clinical experience). All interviews were orally conducted by one researcher (KMR, female dietitian). The results of the study are divided into patient and staff sections. All patients were female, aged 18–27 years (average: 22 years), diagnosed with AN (*n* = 6) and other specified feeding or eating disorders, such as restrictive subtype/atypical anorexia nervosa (*n* = 2). In the patient section, the following factors influenced the acceptability of the nasogastric refeeding protocol: patient’s expectations; awareness and involvement in treatment; communication with the multidisciplinary team; the Mental Health Legislation Act; pros and cons of ‘nil by mouth’; limited individualization; other patients; and uncertainty about the future.

The patients’ answers are given in quoted direct speech.

Most of the patients (*n* = 7) considered the treatment necessary, although one patient experienced it as a punishment. With regard to communication with the medical staff, patients described difficulty in obtaining explanations and clear answers. They expressed the need for further training and education on nasogastric feeding, either in verbal or written form. Patients also reported receiving impersonal treatment and feeling like objects and not people.

The NGT placement was “uncomfortable” and “traumatic” for two participants, while others reported a feeling of relief for being kept dry.

Some patients (*n* = 4) reported to have felt anxious about starting to eat again, while others (*n* = 3) expressed fear at the idea of feeding only with NGT because they worried that they would lose the habit of eating semi-regularly.

In the medical staff section, the following factors influence the acceptability of the nasogastric refeeding protocol for the staff: competence and confidence; working with stigma and ambivalence; working with families; evidence supporting treatment protocol; conflicted feelings about the protocol; and the importance of teamwork.

Almost all (*n* = 11) of them reported that they did not feel ready or comfortable to work with this kind of patient because they were not adequately trained. Only teamwork and comparison with colleagues helped to overcome this lack of confidence. Difficulties were also encountered in establishing working alliances with families, particularly when family members were convinced that the patient could and should eat orally.

The study results therefore emphasized that a patient-centered approach, with clear communication and the combination of a long-term oral diet, is necessary to improve the tolerability of nasogastric feeding. A main limitation is that the representation of participants’ views may be biased, as with all qualitative data analyses. Although efforts were made during interviews to validate interpretations, bias is still a concern. Moreover, the study did not interview family members to gather their perspectives on treatment, despite evidence strongly suggesting their importance in eating disorder treatment. Participants who consented to the study may have represented specific personality types or had experiences within particular treatment contexts, both positive and negative. These have not been objectively assessed and systematically considered in the data analysis, creating a possible interpretation bias. The study was limited to female participants predominantly diagnosed with AN, which may not fully represent the diversity of eating disorder patients.

The study by Fuller et al. (2023) [27] aimed at describing the use of the NGT in severe AN patients with psychiatric comorbidity and evidenced the possible addiction that this physical device may create in them. The study was conducted on a sample of 143 patients, with an average age of 19.02 years (SD = 7.9), and the sample consisted of 77.6% females, 1.4% males (1.4%), and 21% of a gender different from that they were born with, and the main diagnosis was AN (68.5%), whereas depression, anxiety and autism spectrum disorder were the most frequent comorbidities. This case series highlights that even after the period of medical stabilization, there is still a need to use NGT because it likely makes weight gain more tolerable and less difficult in some patients with severe and persistent eating disorders. This case series highlights the chance that some patients may develop an addiction to the NGT. This issue has also been noted in previous studies. This study, however, has some limits: the study excluded settings where NGT feeding under physical restraint might occur outside of inpatient mental health units, such as acute medical or pediatric wards, which could limit the generalizability of the findings. Moreover, the study focused on adolescent inpatient units, which limits its applicability to adult populations. It did not survey general adult inpatient units, so the potential use of NGT feeding under restraint for adults with conditions like obsessive–compulsive disorder or emotional dysregulation with food refusal was not explored. These limitations highlight potential biases, challenges in data accuracy, and gaps in generalizability to broader clinical settings and adult populations. No standardized assessment of patients’ perception of NGT treatment was carried out.

### 3.2. General Comments on the Studies

The six selected studies [22,23,24,25,26,27] evidence that NGT may represent a useful device accepted by the majority of AN patients when other treatments (e.g., CBT) are not sufficient to reduce their fear of being nourished. Sometimes it may also be imposed with physical restraint, even though this application produces management issues, which may be sometimes overcome using psychotropic drugs, in particular in adults. The psychological effects of NGT use may range from relief of the anxiety of eating to the stoic acceptance of its need for treatment, to addiction and fear of restarting oral eating. A new issue that emerged from the reviewed literature was the ambivalence towards NGT by the staff who deserve specific training on its management and the management of parents.

The analysis of the studies evidences some discrepancies in the level of tolerability of the NGT. Even if the majority of the studies underline the feasibility of the NGT treatment, some of them suggest that drug treatment may be necessary for helping patients’ cooperation [22,25], while the study of Blikshavn et al. [24] suggests that especially in the minor-age population, restraint may be a better option to avoid medications’ side effects. This may depend on the different patient populations and treatment settings that have been considered, ranging from pediatric wards with cooperative patients to psychiatric wards hosting severely affected psychiatric patients. Also, the staff’s ability and attitude [26] in motivating and supporting patients and their families may have had a role in the management difficulties.

Some relevant biases emerge in the studies’ methodology. First off, the report of selected cases may produce either a positive or a negative bias with respect to patients’ and authors’ opinions toward NGT use. The sex differences among recruited samples (mainly female but in one case exclusively male) produce a significant selection bias since AN displays different features in the two sexes. Moreover, the low number of included cases, which characterize the majority of the included studies, reduces the representativeness of the results for the whole AN population. Finally, the methodological differences among the studies (including the non-objective assessment of psychological perception of NGT use, the non-systematic assessment of psychiatric comorbidity, and eating and general psychopathology features) impairs the comparability of their results.

Notwithstanding these apparent limitations, the inclusion of studies applying different methodologies granted the possibility of some new findings with respect to previous literature enlarging the points of view on NGT use from the mere tolerance to issues concerning its management (e.g., with combined drug treatment or contention) and the need for specific training of the staff.

### 3.3. Quality of Studies

Two independent reviewers analyzed the trials and assessed some comparable aspects that gave information on quality assessment listed in Table 3.

While the aims of the studies were generally clear and the study design appropriate, the sample size of the six included studies was adequate to represent the AN population only in two studies. Moreover, only half of the studies represent a cohort study, while the other three were case series; thus, they included a good qualitative analysis but without consistent statistics on quantitative data. Finally, only one was a controlled randomized trial, granting sufficient strength to the findings.

## 4. Discussion

The present review collected only six studies published since the last review on the topic of psychological effects of NGT in AN, i.e., in the last 14 years. The use of NGT in the treatment of this disorder greatly increased in the post-COVID pandemic because of the absolute increase and the relative worsening of the psychopathology of these disorders [4,5]. Thus, it is very peculiar that so few studies were conducted in such a long time concerning the psychological effects of a such widely used instrument for the care of the AN. This lack of research is even more relevant considering that AN is eminently a psychological disorder and that the previous literature did not explore in sufficient depth its short-term and long-term consequences on the psychopathological evolution of the AN subjects [19]. Thus, the aim of the current review was to highlight the new findings and challenges proposed by research on this topic in the last 14 years and underline what is still lacking to support future research.

### 4.1. Study Characteristics

Among the studies included in the present review, only one aimed to assess psychological aspects in patients with NGT, while the other studies had different objectives, e.g., to identify the clinical characteristics of patients receiving NGT feeding under physical restraint. However, since the studies included the description of patients’ experiences during treatment, relevant information on the psychological tolerability of the NGT perceived by the participants could be extrapolated. For this reason, studies with different research purposes were also included in our review. The overall quality of the data recorded from the included studies is not statistically strong because the only study specifically assessing the psychological effects of NGT administration [22] was a qualitative one reporting on direct patient talk through interviews. This justifies the possibility of a review and not of a meta-analysis on the topic. Moreover, in addition to the methodological limitations concerning the design of the studies, also the different treatment settings and populations considered in each study and the differences in treatment staff characteristics contributed to relevant discrepancies between the study results. The absence of a systematic assessment of personality and psychopathology, and the subjective perception of the experience with the NGT treatment, reduces the strength and comparability of the findings. In conclusion, the relatively low quality of the data suggests that the findings are only preliminary and need replication with more quantitative and methodologically sound (e.g., RCT) designs.

### 4.2. NGT Tolerability

All the studies that have been analyzed [22,23,24,25,26,27] show that the proposal of the clinician to use the NGT on patients with AN is a source of fear and stress for most patients, parents, and even staff. Their opinion on the NGT, however, changes once it is actually used. In fact, most patients claim to tolerate the use of the nasogastric tube well [18,19,22,28,29], both from a physical and psychological point of view. In fact, NGT feeding may initially be perceived as intrusive or distressing for patients, as they may have the belief that it is a tool that will quickly lead to weight gain; moreover, it does not allow them to know exactly how much food they are ingesting and it may also be a source of stress for parents to see their child with NGT. Possible subjective features influencing the AN patients’ attitudes towards NGT encompass comorbid psychiatric psychopathology (e.g., anxiety or mood disorders), personality traits, or personality disorders, which may influence subjective responses to stressors [5,18]. Also, information retrieved on the internet or by social media, or personal reports by patients who experienced NGT use, may display some influence on individuals’ perception of NGT experience, but to the best of our knowledge, they have not been considered in any study up to now. As suggested by some studies, the use of drugs [22,25] and even physical restraint [30] may sometimes overcome these initial difficulties. Often, however, especially as the weeks go by, it becomes an accepted and even welcome aspect of the treatment, as patients begin to recognize its role in facilitating physical and psychological recovery, reducing the subjective stress consequent with introducing a large amount of food. In fact, with the use of NGT, the patients claim to feel relieved of the guilt of consuming calories. This is possibly due to the reduction in patient’s decision-making stress related to food intake, which raises the conflictual instances between hunger and desire for food and the extreme need to control it.

### 4.3. NGT Treatment Steps

One step of the treatment with NGT was particularly critical: the insertion.

The collection of the consensus for NGT insertion is critical because of the above-described negative prejudices of patients and parents, which may stimulate resistance or the decision to refuse its use. For this reason, some studies report the necessity of restraint tools to ease the insertion of the nasogastric tube against the will of the patient [25,30]. To make the insertion of the NG tube less traumatic for patients, sometimes emotional support provided by experienced staff suffices; however, administering an anxiolytic or atypical antipsychotic before the procedure could also be considered [22,30]. Many doubts concerning the use of restraint arise, as it may not be a good practice in the long term. In fact, it appears that this practice, which forces the will of the patients, could represent a relevant obstacle in establishing a relationship of trust between sanitary staff and the patients. And the situation becomes even harder if parents themselves are asked to help restrain their children [29]. The effect of this forcing could be highly traumatic for both the patient and parents and could cause a sequence of bad moods, misunderstandings and negative feelings [31,32,33]. As an alternative to this practice, the aim to be set is instead a patient-centered and dialectic dialogue between the parties that allows clear communication on the benefits and possible complications of enteral tube feeding to overcome prejudices and irrational fears. Also, underlining the good effects on undesirable eating behaviors, e.g., the 81% versus 38% achieving abstinence from binge-purging behaviors with the NGT use, could encourage patients to change their fears into alliance with treatment [34].

Also, the removal of the tube might be a source of worry. Studies [26,28] report that some patients feel anxiety at the thought of starting to eat orally again. This may be generated both by the need to stop the “nil in the mouth” condition [25], and at the same time to renounce the reassuring condition of being passively fed “as a little baby”, which may satisfy the needs of regressive attachment for these patients [35]. In this case, it is clear that the NGT has to be removed when the nutritional targets have been reached. Nevertheless, as for the former critical point, only the construction of a deep and trustful relationship between the patients and the medical and psychological staff may prevent the psychological problems concerning oral renutrition [36,37].

The NGT can therefore be considered an ambivalent instrument in the hands of medical staff. Its image in the mind of patients may fluctuate in several different forms regarding the phase of the treatment, ranging from an unwanted medical imposition to a necessary, reassuring, and essential tool. As it was claimed by a patient, “NGT made me feel better even though I didn’t want it” [22].

### 4.4. Others’ Points of View: Parents and Medical Staff

The current review also raises the point that it is impossible to talk about NGT without considering two other key actors involved in its use: parents and medical staff.

As evidenced by the studies analyzed in our review [22,23,24,25,26,27], the patients suffering from AN are usually young and often minors, and therefore they are accompanied in the treatment choices by their parents. Moreover, it is well known that the illness of one member can involve the whole family, becoming a collective disease [37,38,39,40]. It is therefore fundamental to involve the parents in the therapeutic path, educating them about the psychopathological functioning, clinical features, and behavioral expressions of the disease and informing them about the different options and phases of the therapeutic process [41,42,43]. The primary objective is to establish a relationship of trust with the family as a whole, reducing the negative feelings of anger, discouragement, fear, and prejudices towards therapeutic tools like the NGT [41,42,43,44]. This may help to trans-motivate patients to cooperate with clinical staff, facilitating the use of necessary therapeutic treatments, even if unpleasant, and also reducing the stress of their interruption.

Only one study by Matthews-Rensch et al. (2022) [26], among those included in the present review, reports the perspective of the sanitary staff. The survey administered to the healthcare personnel permits the collection of the difficulties encountered in working with the patient population with severe eating disorders, describing generalized feelings of “uncomfortableness” both towards patients and their resistances and also with “unpleasant” therapeutic instruments such as the NGT. Also, in this case, the negative perception of the staff eventually becomes better with time. On the other hand, the studies claim the importance of teamwork, the need for discussion with all medical staff, and the need for clear protocols approved in multidisciplinary meetings [26,45,46,47].

Collected studies suggest that NGT feeding, even if faced in the first instance with prejudices and fears, is psychologically well tolerated by patients with AN; nevertheless, many aspects remain underexplored. For instance, patients with AN in the acute phase of the disease often exercise obsessive control over food and calories ingested. The use of NGT does not allow them to maintain such control, and the loss of control may worsen or improve their therapeutic alliance and subjective distress in different ways, depending on their personality, psychopathology, alliance with the therapeutic team, and phase of the disease treatment. Thus, further research is needed to improve the knowledge of the possible consequences of NGT use to better tailor the treatment to a patient’s characteristics.

Another issue that arose from the review is the need for greater involvement of the parents and even the staff in the psychoeducation about strengths and limitations and the possible positive and negative consequences of the NGT use because the motivation or the fear of the caregivers around the patients may heavily influence the patient’s acceptance of the NGT. In particular, training and support for medical staff can improve their confidence and reduce their discomfort when using NGT, or the need for restraints that in some studies overcome 60% of the included patients [22,30]. The mechanism underlying the relevance of this issue is “trans-motivation”, a process by which the staff empathically validates the patients with respect to their fatigue in accepting the NGT treatment, but also suggests to them that their task is sustainable, secure, and will lead to future satisfaction [30]. To produce an effective “trans-motivation”, the staff needs to deeply believe what is asked of the patients, so specific training on the pros and cons of NGT treatment is needed. This training should be supported by new evidence concerning subjective psychological acceptance and psychological benefits of the NGT treatment in AN, other than with respect to the weight targets [16]. Thus, it will be very useful for future studies to produce a more in-depth analysis of the psychological effects of NGT in AN and to extend the examination of psychological implications to families and healthcare staff since their role in the NGT acceptance is essential.

### 4.5. Recommendations for Further Research

The literature on NGT use for AN in the last 14 years evidences a severe lack of knowledge regarding the psychological consequences of its application. The psychological prejudices of patients, families, and staff before the introduction of NGT and the immediate psychological effects after the introduction have not been systematically assessed with specific assessment instruments, which should be prepared and validated for future research. Also, the quantitative assessment of the evolution of the psychological attitudes of the patients regarding the NGT treatment is completely lacking and should be carried out in prospective studies. Moreover, patients’ personality or psychopathology features, which may influence the acceptance or the perception of side-effects of this treatment, have not been explored with objective instruments and were not considered in the statistical analysis of the studies as possible confounders (where it was performed). Future research should take these features into account as influential factors in the management and evolution of the NGT treatment. There is also a lack of research on the medium- and long-term evolution of the NGT treatment after its interruption. Finally, a clear definition of the recruitment settings and their possible influence on the treatment outcome needs to be considered in future research. Thus, both cross-sectional and longitudinal studies on the long-term psychological effects of NGT would provide clearer guidance for clinicians and may help to produce a more effective motivation of patients, families, and staff in the application of this instrument, when needed. Additionally, proposing specific interventions or training programs to improve communication and trust building could be valuable for practitioners.

### 4.6. Limits of the Study

The main limitation of our review is the small number of included studies and, consequently, the small patient population analyzed. Moreover, the eminent qualitative nature of the reviewed studies, the poorly objective psychological assessment of patients, staff, and parents, and the small number of included subjects suggest that there is a need for more quantitative studies with high-quality data on larger samples of patients to produce more objective and definitive data.

Notwithstanding its limitations, emerging from the current review is a general recommendation for creating clear guidelines on tube-feeding in anorexia patients to reassure staff, patients, and parents about necessary procedures in the use of an instrument, which often looks critical. Moreover, the review also suggested that significant attention should be paid to developing relational and communication skills among therapeutic teams to improve communication among healthcare staff, parents, and patients, with the purpose of making the practice of NGT use as comfortable as possible with the aim of avoiding constraints and physical restraints.

## 5. Conclusions

The present review raises relevant issues in the current research on the use of NGT in AN. First, as can be seen from the low number of selected studies, the general topic of the NGT application to AN treatment is highly overlooked. The author’s opinion is that it is crucial to assess the psychological suffering, stress, or relief that the introduction (and removal) of NGT might cause, because the AN patients’ mental health is in precarious balance and any change in disease-maintaining mechanisms may hinder or foster therapeutic processes. Collected studies suggest that NGT feeding, even if faced in the first instance with prejudices and fears, is psychologically well tolerated by patients with AN; nevertheless, many aspects remain to be explored.

## Figures and Tables

**Figure 1 nutrients-16-02316-f001:**
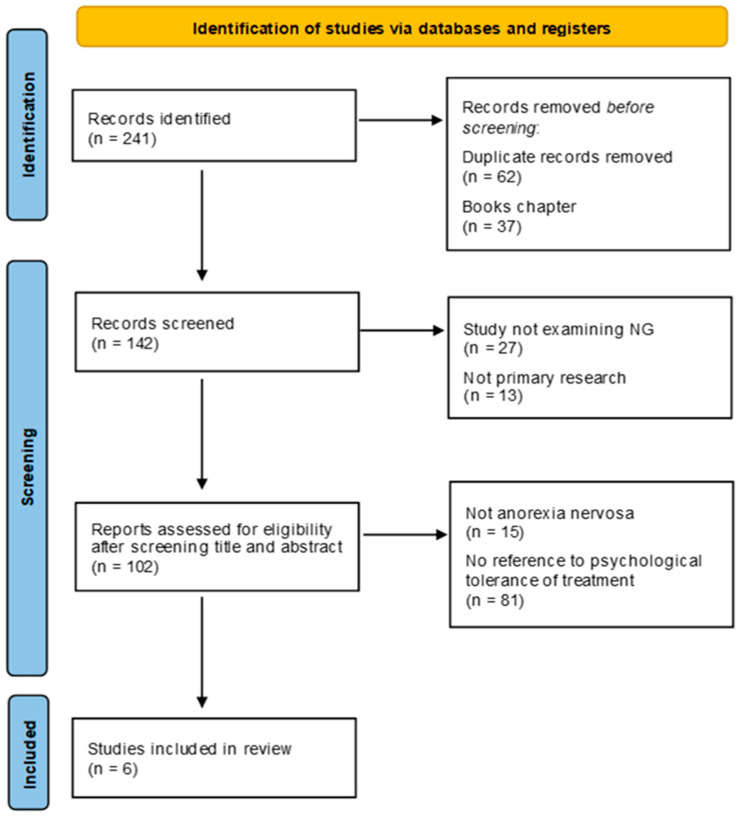
PRISMA 2020 flow diagram for new systematic reviews, which included searches of databases and registers only.

**Table 1 nutrients-16-02316-t001:** Summary of the main characteristics of the included studies.

Study	Design	Country	Setting	Aims
Bayes and Madden (2011) [22]	Retrospective case series	Australia	Hospital	To describe the demographic and clinical features of male inpatients with early-onset eating disorders
Rigaud et al. (2011) [23]	Prospective cohort study	France	Outpatient service	Abstinence from binge eating/purging episodes, improvements in nutritional status, psychological and quality of life Improvements
Blikshavn et al. (2020) [24]	Quantitative study of follow-up	Norway	Regional, specialized adolescent eating disorders inpatient unit offering a family-based inpatient treatment	To describe the frequency of physical restraint in a specialized program for adolescents with AN, and to examine if meal-related physical restraint (forced nasogastric tube feeding) was related to 5-year outcome
Falcoski et al. (2021) [25]	Case series	UK	Hospital (specialist eating disorders unit for children and adolescents)	To illustrate practices in line with new dietetic guidelines for NGT feeding under restraint
Matthews-Rensch et al. (2022) [26]	Qualitative exploratory study	Australia	Tertiary hospital	To describe the acceptability of a nasogastric refeeding protocol with adult patients with medically unstable eating disorders and the staff involved in their treatment
Fuller et al. (2023) [27]	Comprehensive audit and case series	UK	n.a.	To identify the clinical characteristics of patients receiving nasogastric tube (NGT) feeding under physical restraint

n.a. = no available data.

**Table 2 nutrients-16-02316-t002:** Main data of included patients.

Study	Sample	Age in Years	Gender	Length Stay (Average)	Diagnosis
Bayes and Madden (2011) [22]	10	12.8 (10.6–14.5)	Male (100%)	36.3 days	AN (30%) EDNOS (70%)
Rigaud et al. (2011) [23]	103	27.4 (19.3–35.5)	Female (100%)	n.a.	AN (35%) BN (65%)
Blikshavn et al. [24]	38	15.9 (SD = 1.9)	Female (89.5%) Male (10.5%)	20.3	AN (100%)
Falcoski et al. [25]	3	11 (33%) 14 (67%)	Female (67%) Male (33%)	n.a.	AN (67%) Atypical AN (33%)
Matthews-Rensch et al. (2022) [26]	8	22 (18–27)	Female (100%)	n.a.	AN (75%) Atypical AN (12.5%) OSFED (12.5%)
Fuller et al. (2023) [27]	143	19.02 (SD = 7.9)	Females (77.6%) Males (1.4%) Gender different from that they were born with (21%)	29.1 weeks	AN (68.5–75.7%) BN (0.7–3.2%) OSFED (6.7–9.1%) Others

n.a. = no available data.

**Table 3 nutrients-16-02316-t003:** Study quality data.

Study	Aims	Sample Size	Justified Sample Size?	Level of Evidence
Bayes and Madden (2011) [22]	Aims are clear and the study design is appropriate.	10	The study sample is too small and not representative of the reference population.	Case series
Rigaud et al. (2011) [23]	Aims are clear and the study design is appropriate.	103	Sample size is suitable.	Cohort study
Blikshavn et al. (2020) [24]	Aims are clear and the study design is appropriate.	38	The study sample is too small and not representative of the reference population.	Cohort study
Falcoski et al. (2021) [25]	Aims are clear and the study design is appropriate.	3	The study sample is too small and not representative of the reference population.	Case series
Matthews-Rensch et al. (2022) [26]	Aims are clear and the study design is appropriate.	8	The study sample is too small and not representative of the reference population.	Cohort study
Fuller et al. (2023) [27]	Aims are clear and the study design is appropriate.	143	Sample size is suitable.	Case series

## Data Availability

Data are available upon reasonable request; the lead authors have full access to research data.

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
