# Peer review of "Psychological Effects of Nasogastric Tube (NGT) in Patients with Anorexia Nervosa: A Systematic Review"

_nutrients, 2024, doi:10.3390/nu16142316_

Round 1

Reviewer 1 Report

Comments and Suggestions for Authors

This study highlights the psychological aspect of nasogastric tube (NGT) use in patients with anorexia nervosa

1. Detailed flow of the  study selection process incluidng PICO

2. Discussion section

 address the limitations of this studies and review : I think this is addressed in conclusion section -> move it

3. Conclusion

  too long, rearrage the findings or summary  to results or disscusstion

 provide a brief and concise conclusion based on the results

Author Response

Reviewer 1

This study highlights the psychological aspect of nasogastric tube (NGT) use in patients with anorexia nervosa.

  1. Detailed flow of the  study selection process incluidng PICO

The authors specified in the methods section that “The research question for the study was developed following the PICO (Population, Indicator, Comparator, and Outcome) criteria to create an effective research strategy.” The authors added these two parargraphs:

2.4. Data Extraction

The research questions and data extraction were guided by the PICO framework. The study population included all patients with anorexia nervosa according to DSM5 criteria, fed by nasogastric tube. There were no age or gender restrictions. The intervention we analyzed was enteral feeding via nasogastric tube, and our primary outcome measure was the psychological tolerance of the patients.  The secondary outcome measure was the impact of this feeding practice on the healthcare team and carers.

2.5. Quality Assessment

Due to the observational design of the majority of the studies analyzed, it was not possible to use a formal evaluation instrument [46]. The limitations of each study are described in the next section.

Moreover, in the methods section it was added a description of the figure 1 in which the authors detail the flow of selection process.

  1. Discussion section address the limitations of this studies and review : I think this is addressed in conclusion section -> move it

As suggested by the reviewer the authors moved the limitations of the study from the conclusions to the  discussion section.

  1. Conclusion  too long, rearrage the findings or summary  to results or disscusstion provide a brief and concise conclusion based on the results

The conclusion section have been rearranged and summerized as requested by the reviewer.

Reviewer 2 Report

Comments and Suggestions for Authors

Abstract:

·         The aim is stated clearly, emphasizing the increased need for intensive nutritional care in anorexia nervosa (AN) patients post-COVID-19 and exploring the psychological effects of NGT feeding. This contextualizes the relevance of the study well. However, specifying that this is a systematic review in the aim statement would strengthen the abstract.

·         The results section effectively summarizes the findings, mentioning the total number of studies found and the number excluded. It is noted that only five studies met the inclusion criteria, and these included various study designs. The abstract could benefit from briefly mentioning the main findings from these studies, particularly the observed psychological effects.

1. Introduction:

·         The introduction of NGT as a necessary intervention for patients in acute phases or those non-compliant with dietary therapy is logically presented. However, the explanation could benefit from more detail on the specific circumstances under which NGT is deemed necessary.

·         Mentioning previous studies that have explored the clinical benefits of NGT but not its psychological impacts sets the stage for the study’s unique contribution.

·         The review of studies preceding 2010 provides historical context, but more recent studies (post-2010) are referenced sparingly. A more balanced discussion incorporating both older and more contemporary research would strengthen the literature review.

·         The introduction could benefit from a more critical evaluation of the methodologies and findings of these earlier studies to better highlight the existing knowledge gaps.

·         The introduction flows logically from defining AN, discussing its prevalence, and explaining the necessity of NGT, to identifying the research gap. This clear structure aids in reader comprehension. However, some sentences are overly complex and could be simplified for clarity. For example, “The treatment of AN is complex and multidisciplinary encompassing psychiatric, nutritional and psychological approaches” could be broken down into shorter, more digestible statements.

2. Review Methodology

·         Including various databases such as WorldCat, MEDLINE®/PubMed®, Scopus, Directory of Open Access Journals, and Wiley Online Library is commendable. However, the methodology would benefit from a more detailed explanation of why these databases were chosen and if any were considered but excluded.

·         The keywords and MeSH terms used are appropriate, but a more comprehensive list of search terms could ensure broader coverage of the literature. Additionally, explaining how these terms were combined and any Boolean operators used would enhance reproducibility.

·         The criteria set for inclusion and exclusion are well-defined. However, the rationale behind not limiting the search to English-language articles is not explicitly stated. While this inclusivity benefits a comprehensive review, it might pose challenges in translating and interpreting non-English studies.

·         The decision to exclude reviews is reasonable, yet it would be useful to justify this choice, considering reviews can offer valuable summarized insights.

3. Results:

·         The section is well-organized, but the flow of information could be improved. For instance, the detailed description of the studies could be presented more cohesively, possibly integrating the key findings and their implications within the text rather than separating them.

·         The tables provided (Table 1 and Table 2) are useful but could be more comprehensive. Including additional details such as the sample size, specific psychological outcomes measured, and statistical significance would provide a clearer picture of the results.

·         The description of each study is detailed, but it lacks a critical assessment of the methodological strengths and weaknesses. For example, the study designs are mentioned (e.g., case series, qualitative exploratory study), but there is no discussion on the potential biases or limitations inherent in these designs.

·         The inclusion of studies with different designs is justified, but a more detailed rationale for their inclusion despite varying objectives would strengthen the review. It would be helpful to explain why studies with different primary aims were included and how their findings contribute to the overall understanding of the psychological effects of NGT.

·         The interpretation of the results is somewhat superficial. While the section mentions that patients generally tolerate NGT well, there is limited discussion on the potential variability in psychological responses. For example, it would be beneficial to explore why some patients may experience NGT as traumatic while others find it helpful.

·         There is a mention of the "relief" some patients feel due to NGT, but this could be expanded upon. Discussing the psychological mechanisms behind this relief, such as reducing decision-making stress related to food intake, would provide deeper insights.

·         The critical analysis of the included studies is lacking. There is no discussion on the quality of the studies or the robustness of their findings. Incorporating a critical appraisal of each study’s methodology, including sample size, study duration, and data collection methods, would enhance the review's credibility.

·         The manuscript would benefit from a more critical comparison of the studies. Highlighting discrepancies in findings and discussing potential reasons for these differences (e.g., cultural factors, differences in NGT protocols) would add depth to the analysis.

4. Discussion:

·         The discussion could benefit from a more detailed analysis of the methodological strengths and weaknesses of the included studies. For example, discussing the sample sizes, study designs, and potential biases in more depth would provide a clearer picture of the robustness of the findings.

·         Including more specific data or direct quotes from the studies to illustrate these points would be beneficial. Additionally, discussing the potential factors contributing to this change in perception (e.g., improved physical health, and psychological support) would add depth to the analysis.

·         This section could be strengthened by integrating more evidence from the included studies about the specific challenges and strategies used during these critical steps. For example, detailing successful approaches to obtaining consent or managing anxiety during tube removal would be useful for practitioners.

·         While the perspectives of patients and parents are covered, the section on medical staff could be expanded. It would be beneficial to discuss how training and support for medical staff can improve their confidence and reduce their discomfort when using NGT. Additionally, exploring how the perspectives of parents and staff influence patient outcomes could provide a more integrated view.

·         The recommendations could be more specific. For instance, suggesting concrete areas for future research, such as longitudinal studies on the long-term psychological effects of NGT, would provide clearer guidance for researchers. Additionally, proposing specific interventions or training programs to improve communication and trust-building could be valuable for practitioners.

Comments on the Quality of English Language

The quality of the English in the manuscript is generally clear and understandable. However, it contains several minor grammatical errors, awkward phrasing, and inconsistencies that require attention. 

Author Response

Answer to the reviewers

The authors thank the reviewer for the careful reading and revision of the manuscript and for the suggestions that improved it for publication.

Here follow the detailed answers (in blue) to reviewer’s comments, the authors have tracked the text changes with blue characters.

Reviewer 2

Abstract:

  • The aim is stated clearly, emphasizing the increased need for intensive nutritional care in anorexia nervosa (AN) patients post-COVID-19 and exploring the psychological effects of NGT feeding. This contextualizes the relevance of the study well. However, specifying that this is a systematic review in the aim statement would strengthen the abstract.

The authors have now specified that this is a systematic review in the abstract.

  • The results section effectively summarizes the findings, mentioning the total number of studies found and the number excluded. It is noted that only five studies met the inclusion criteria, and these included various study designs. The abstract could benefit from briefly mentioning the main findings from these studies, particularly the observed psychological effects.

The authors now mentioned in the abstract the main findings for the included studies:

“Included studies suggest that NGT feeding, even if faced in a first instance with prejudices and fears both by patients and staff, is psychologically well tolerated by patients with AN. Nevertheless, recent in-depth research on the issue is lacking, the existing has a low methodological quality, thus many psychological effects of NGT application remain underexplored. “

  1. Introduction:
  • The introduction of NGT as a necessary intervention for patients in acute phases or those non-compliant with dietary therapy is logically presented. However, the explanation could benefit from more detail on the specific circumstances under which NGT is deemed necessary.

The authors now specified with greater detail the circumstances in which the NGT use may be requested for the cure of AN patients:

The introduction of NGT in patients with anorexia nervosa (AN) is typically considered in specific clinical scenarios [10, 11]:

  1. Severe Malnutrition and Weight Loss: When patients present with severe malnutrition or have lost a significant amount of weight (typically more than 15-20% of their body weight), nutritional support through an NGT may be necessary to prevent further health deterioration and to stabilize the patient’s condition. This is especially critical when oral intake is insufficient to meet the caloric and nutritional needs required for recovery (Hay et al., 2014).
  2. Failure of Oral Refeeding: If patients are non-compliant with dietary therapy or oral refeeding plans, an NGT can ensure that they receive the necessary nutrients. This approach helps in overcoming the resistance to eating commonly observed in AN, thereby facilitating weight gain and nutritional rehabilitation (American Psychiatric Association, 2020).
  3. Critical Clinical Condition: In cases where patients are in a life-threatening condition due to complications such as electrolyte imbalances, cardiovascular instability, or severe organ dysfunction, immediate nutritional intervention via NGT is crucial to support vital functions and to initiate the recovery process. (Royal College of Psychiatrists, 2017).

Overall, the decision to introduce an NGT is made on a case-by-case basis, taking into account the patient’s medical condition, nutritional status, and the risk-benefit ratio of the procedure. The goal is to stabilize the patient, ensure adequate nutrition, and transition back to oral feeding as their condition improves.  [9, 12].

  • Mentioning previous studies that have explored the clinical benefits of NGT but not its psychological impacts sets the stage for the study’s unique contribution.

The authors have now added some citations describing the current “state of art” of NGT supplementation in the treatment of AN patients.

  • The review of studies preceding 2010 provides historical context, but more recent studies (post-2010) are referenced sparingly. A more balanced discussion incorporating both older and more contemporary research would strengthen the literature review.

Now the authors added some more citations of post-2010 studies to incorporate more recent findings on the topic.

  • The introduction could benefit from a more critical evaluation of the methodologies and findings of these earlier studies to better highlight the existing knowledge gaps.

The authors now provided a more critical evaluation of the methodologies and of the findings of the earlier studies to highlight the existing knowledge gaps.

  • The introduction flows logically from defining AN, discussing its prevalence, and explaining the necessity of NGT, to identifying the research gap. This clear structure aids in reader comprehension. However, some sentences are overly complex and could be simplified for clarity. For example, “The treatment of AN is complex and multidisciplinary encompassing psychiatric, nutritional and psychological approaches” could be broken down into shorter, more digestible statements.

Long sentences have been revised according to the reviewer's comment.

  1. Review Methodology
  • Including various databases such as WorldCat, MEDLINE®/PubMed®, Scopus, Directory of Open Access Journals, and Wiley Online Library is commendable. However, the methodology would benefit from a more detailed explanation of why these databases were chosen and if any were considered but excluded.

The authors now specified that “the included databases were all those available online for international medical literature research, none was excluded”

  • The keywords and MeSH terms used are appropriate, but a more comprehensive list of search terms could ensure broader coverage of the literature. Additionally, explaining how these terms were combined and any Boolean operators used would enhance reproducibility.

The authors used the search terms which were focused on the specific issue of the review, they were not included “psychological” key words to include the highest possible number of studies that were successively explored by the authors to focus the review.

The authors now specified that:

The keywords used combined NGT and anorexia nervosa; nasogastric tube and anorexia nervosa; tube feeding and anorexia nervosa; enteral feeding and anorexia nervosa.

  • The criteria set for inclusion and exclusion are well-defined. However, the rationale behind not limiting the search to English-language articles is not explicitly stated. While this inclusivity benefits a comprehensive review, it might pose challenges in translating and interpreting non-English studies.

The authors have choosen not to select the studies for language because of the paucity of the findings evidenced by preliminar analysis. We now specified it in the text:

We chose not to narrow the search due to the scarcity of articles on the topic.

We therefore decided not to limit the search to English language articles, given the limited number of studies available in the literature on this topic. It also didn't impose any restrictions on the type of study with the sole exclusion of reviews.

  • The decision to exclude reviews is reasonable, yet it would be useful to justify this choice, considering reviews can offer valuable summarized insights.

The authors agree with the reviewer, nonetheless the current opinion of researches performing the reviews suggest not to include reviews in a systematic review to improve the originality of the findings and respect the temporal limits of the review.

  1. Results:
  • The section is well-organized, but the flow of information could be improved. For instance, the detailed description of the studies could be presented more cohesively, possibly integrating the key findings and their implications within the text rather than separating them.

The authors preferred this exposition method because of the scarce number of the studies and of the poor coherence of their methodology and findings, which could be thus poorly integrated. Moreover the common findings and the implications are discussed in the discussion section. The authors adopted summary tables to evidence comparable characteristic. Finally, an overall report on studies methodological features have been added.

  • The tables provided (Table 1 and Table 2) are useful but could be more comprehensive. Including additional details such as the sample size, specific psychological outcomes measured, and statistical significance would provide a clearer picture of the results.

The requested details have been included where available.

  • The description of each study is detailed, but it lacks a critical assessment of the methodological strengths and weaknesses. For example, the study designs are mentioned (e.g., case series, qualitative exploratory study), but there is no discussion on the potential biases or limitations inherent in these designs.

A table reporting of the limitations of each study was now added (Table 3), and the results summarized in the pertaining paragraph.

  • The inclusion of studies with different designs is justified, but a more detailed rationale for their inclusion despite varying objectives would strengthen the review. It would be helpful to explain why studies with different primary aims were included and how their findings contribute to the overall understanding of the psychological effects of NGT.

The authors considered reviewer’s comment including this sentence:

The choice of including studies with different primary aims and methodological approach is justified by the scarce number of available research on the topic but also because of their possible contribution in the definition of psychological correlates of the NGT use in AN treatment from different perspectives.

  • The interpretation of the results is somewhat superficial. While the section mentions that patients generally tolerate NGT well, there is limited discussion on the potential variability in psychological responses. For example, it would be beneficial to explore why some patients may experience NGT as traumatic while others find it helpful.

Unfortunately the reviewed studies do not give data supporting a more in-depth discussion of the psychological perception of patients, now the authors provided some reasonable hypothesis.

“Possible subjective features influencing the AN patients’ attitudes towards NGT encompass comorbid psychiatric psychopathology (e.g. anxiety or mood disorders), personality traits or personality disorders which may influence subjective responses to stressors [5]. Also, information retrieved on the internet or by social media, or personal reports by patients who experienced the NGT use may display some influence on individuals’ perception of NGT experience, but to the best of our knowledge they have not been considered in any study up to now”. 

  • There is a mention of the "relief" some patients feel due to NGT, but this could be expanded upon. Discussing the psychological mechanisms behind this relief, such as reducing decision-making stress related to food intake, would provide deeper insights.

As for the previous point also in this case the studies do not provide sufficient information and in-depth exploration, thus any comment on it is speculative. The authors added this comment:

In fact, with the use of NGT the patients claim to feel relieved of the guilt of consuming calories. This is possibly due to the reduction of patient’s decision-making stress related to food intake which raises the conflictual instances between hunger and desire for food and the extreme need to control it.

  • The critical analysis of the included studies is lacking. There is no discussion on the quality of the studies or the robustness of their findings. Incorporating a critical appraisal of each study’s methodology, including sample size, study duration, and data collection methods, would enhance the review's credibility.
  • The manuscript would benefit from a more critical comparison of the studies. Highlighting discrepancies in findings and discussing potential reasons for these differences (e.g., cultural factors, differences in NGT protocols) would add depth to the analysis.

The authors now added a critical report of the methodology of the included studies.

  1. Discussion:
  • The discussion could benefit from a more detailed analysis of the methodological strengths and weaknesses of the included studies. For example, discussing the sample sizes, study designs, and potential biases in more depth would provide a clearer picture of the robustness of the findings.

The authors now added a critical discussion of the methodology of the included studies.

  • Including more specific data or direct quotes from the studies to illustrate these points would be beneficial. Additionally, discussing the potential factors contributing to this change in perception (e.g., improved physical health, and psychological support) would add depth to the analysis.

As mentioned before, the data reported by the studies supporting the reasons for changes in perception are few. Thus the authors included direct quotes from the included studies to add depth to the analysis.

E.g. 1.the need to stop the “nil in the mouth” condition [25]; 2. being passively fed “as a little baby” which may satisfy the need of regressive attachment of these patients [33]; 3. As it was claimed by a patient “NGT made me feel better even though I didn’t want it.” [20]; 4. describing generalized feelings of “uncomfortableness” both towards patients and their resistances and also with “unpleasant” therapeutic instruments such as the NGT

  • This section could be strengthened by integrating more evidence from the included studies about the specific challenges and strategies used during these critical steps. For example, detailing successful approaches to obtaining consent or managing anxiety during tube removal would be useful for practitioners.

Unfortunately the studies do not report these strategies in detail, so any interpretation would be speculative. The poor comments on this issue have been reported.

  • While the perspectives of patients and parents are covered, the section on medical staff could be expanded. It would be beneficial to discuss how training and support for medical staff can improve their confidence and reduce their discomfort when using NGT. Additionally, exploring how the perspectives of parents and staff influence patient outcomes could provide a more integrated view.

 This comment have been added to the section concerning the medical staff:

In particular, training and support for medical staff can improve their confidence and reduce their discomfort when using NGT. This training should be supported by new evidence concerning subjective psychological acceptance and psychological benefits of the NGT treatment in AN, other than with respect to the weight targets [16]. Thus, it will be very useful for future studies to produce a more in-depth analysis of the psychological effects of NGT in AN and to extend the examination of psychological implications to families and healthcare staff since their role in the NGT acceptance is essential.

  • The recommendations could be more specific. For instance, suggesting concrete areas for future research, such as longitudinal studies on the long-term psychological effects of NGT, would provide clearer guidance for researchers. Additionally, proposing specific interventions or training programs to improve communication and trust-building could be valuable for practitioners.

The following chapter has been added at the end of the discussion

4.5 Recommendations for further research

The literature on NGT use for AN in the last 14 years evidences a severe lack of knowledge with respect to the psychological consequences of its application. The psychological prejudices before the introduction of NGT and the immediate psychological effects after the introduction have not been systematically assessed. Also, the quantitative assessment of the evolution of the psychological attitudes of the patients with respect to the NGT treatment is completely lacking. Moreover, patients’ personality or psychopathology features which may influence the acceptance or the perception of side-effects of this treatment have not been explored. They are also lacking research on the medium and long-term evolution of the NGT treatment after its interruption. Thus, both cross-sectional and longitudinal studies on the long-term psychological effects of NGT would provide clearer guidance for clinicians and may help to produce a more effective motivation of patients, families and staff in the application of this instrument, when needed. Additionally, proposing specific interventions or training programs to improve communication and trust-building could be valuable for practitioners.

Comments on the Quality of English Language

The quality of the English in the manuscript is generally clear and understandable. However, it contains several minor grammatical errors, awkward phrasing, and inconsistencies that require attention. 

The authors revised the language of the study to eliminate grammatical errors and other inconsistencies.

Round 2

Reviewer 2 Report

Comments and Suggestions for Authors

Abstract

·         Although the abstract summarizes the number of studies found and excluded, it still lacks a concise summary of the main findings from the included studies main findings, particularly the observed psychological effects.

Introduction

·         While more recent studies are referenced, the discussion could better integrate these with older studies to provide a more comprehensive overview.

·         Some sentences, though improved, remain complex and could be further simplified for better readability.

Review Methodology

·         While the reasons for choosing specific databases are mentioned, the rationale for excluding others is still unclear.

·         The explanation of how search terms were combined is included, but it could be more detailed to enhance reproducibility further.

·         The challenges posed by including non-English articles, such as translation and interpretation issues, are acknowledged but not addressed in detail.

·         Although the decision to exclude reviews is justified, the potential value of reviews in offering summarized insights is not fully explored.

Results

·         Despite improvements, the flow of information could still be more cohesive. Key findings and implications could be better integrated within the text.

·         The critical assessment of methodological strengths and weaknesses is present but could be more thorough, particularly in discussing biases.

·         While the rationale for including studies with different designs is better explained, it could still be more detailed.

·         The interpretation of results could be deeper, especially when exploring variability in psychological responses.

·         The discussion on patient relief mechanisms could further expand to provide deeper insights.

·         The critical appraisal of each study’s methodology is present but could be more comprehensive.

·         The comparison of studies could be more critical, highlighting discrepancies and discussing potential reasons in greater detail.

Discussion

·         The discussion includes methodological analysis but could be more detailed and in-depth.

·         Integrating specific data or direct quotes is improved but could be more extensive.

·         The section covers specific challenges and strategies but could provide more detailed insights.

·         The expanded discussion on medical staff perspectives could delve deeper into how training and support influence outcomes.

·         While more specific, the recommendations could provide clearer guidance for future research and practical interventions.

Comments on the Quality of English Language

The text is generally comprehensible but contains numerous grammatical errors, awkward phrasings, and occasional lack of clarity that require attention to improve readability and coherence. For instance, sentences like "We therefore decided not to limit the search to English language articles. It also didn't impose any restrictions on the type of study with the sole exclusion of reviews." need revision for smoother flow and better clarity. Similarly, the manuscript has minor punctuation issues, terminology consistency, and sentence structure. It would benefit from thorough proofreading and editing to improve the document, correct these issues, and enhance overall readability.

Author Response

Dear reviewer 2,

the authors greatly appreciated your renewed effort in providing further comments that significantly improved our paper. Here follow the answers to your comments that have been evidenced in blue characters.

Comments and Suggestions for Authors

Abstract

  • Although the abstract summarizes the number of studies found and excluded, it still lacks a concise summary of the main findings from the included studies main findings, particularly the observed psychological effects.

Dear referee the authors better evidenced the findings as follows:

Included studies suggest that NGT feeding, even if faced in a first instance with prejudices and fears by patients, parents and staff, is useful not only for weight increase in treatment-resistant patients with AN, but it also alleviates their stress for feeding and in general it is psychologically well tolerated.

Introduction

  • While more recent studies are referenced, the discussion could better integrate these with older studies to provide a more comprehensive overview.

The authors provided summaries of the previous studies on nutritional and psychological consequences of the NGT.

  • Some sentences, though improved, remain complex and could be further simplified for better readability.

The language was re-revised according to the reviewer’s suggestions.

Review Methodology

  • While the reasons for choosing specific databases are mentioned, the rationale for excluding others is still unclear.

Existing databases which may be excluded were not available on our academic research instrument (EUREKA).

  • The explanation of how search terms were combined is included, but it could be more detailed to enhance reproducibility further.

The authors now specified as follows:

The keywords used were combined as follows: a. NGT and anorexia nervosa; b. nasogastric tube and anorexia nervosa; c. tube feeding and anorexia nervosa; d. enteral feeding and anorexia nervosa.

  • The challenges posed by including non-English articles, such as translation and interpretation issues, are acknowledged but not addressed in detail.

The authors included this comment:

The authors are aware of the possible difficulties derived from the need for translation and/or interpretation of the relevant non-English articles, but since payment translation services and free translation services based on AI are available they accepted the challenge with the aim of producing a complete review.

  • Although the decision to exclude reviews is justified, the potential value of reviews in offering summarized insights is not fully explored.

The authors now commented on the potential value of reviews in offering summarized insights.

Results

  • Despite improvements, the flow of information could still be more cohesive. Key findings and implications could be better integrated within the text.

The authors revised the findings to produce a better cohesion, also evidencing findings and implications.

  • The critical assessment of methodological strengths and weaknesses is present but could be more thorough, particularly in discussing biases.

Some more discussion concerning biases was added.

  • While the rationale for including studies with different designs is better explained, it could still be more detailed.

Some more comments on the rationale have been added after the systematic analysis of the studies.

  • The interpretation of results could be deeper, especially when exploring variability in psychological responses.

It was written a new section including general comments (3.2. section) on psychological responses to deepen the studies interpretation

  • The discussion on patient relief mechanisms could further expand to provide deeper insights.

This issue was expanded in the discussion section

  • The critical appraisal of each study’s methodology is present but could be more comprehensive.

Some more issues have been added to comment the methodology of each study

  • The comparison of studies could be more critical, highlighting discrepancies and discussing potential reasons in greater detail.

3.2. section now comprehensively commented the general findings and the limitations of the studies and their discrepancies

Discussion

  • The discussion includes methodological analysis but could be more detailed and in-depth.

Methodological discussion of the studies was added in the 3.2. new paragraph. A summary of these comments was included in the discussion section.

  • Integrating specific data or direct quotes is improved but could be more extensive.

Some more data have been integrated in the discussion.

  • The section covers specific challenges and strategies but could provide more detailed insights.

Some more detailed insights have been added

  • The expanded discussion on medical staff perspectives could delve deeper into how training and support influence outcomes.

 A more detailed description of the mechanism was given

  • While more specific, the recommendations could provide clearer guidance for future research and practical interventions.

More specific recommendations have been added in the discussion

Comments on the Quality of English Language

The text is generally comprehensible but contains numerous grammatical errors, awkward phrasings, and occasional lack of clarity that require attention to improve readability and coherence. For instance, sentences like "We therefore decided not to limit the search to English language articles. It also didn't impose any restrictions on the type of study with the sole exclusion of reviews." need revision for smoother flow and better clarity. Similarly, the manuscript has minor punctuation issues, terminology consistency, and sentence structure. It would benefit from thorough proofreading and editing to improve the document, correct these issues, and enhance overall readability.

The paper has been revised according to reviewers’ requests.